# Endothelial Dysfunction Syndromes after Allogeneic Stem Cell Transplantation

**DOI:** 10.3390/cancers15030680

**Published:** 2023-01-22

**Authors:** Dionysios Vythoulkas, Panagiotis Tsirigotis, Marianna Griniezaki, Ioannis Konstantellos, Ioanna Lazana

**Affiliations:** 2nd Department of Internal Medicine, Propaedeutic, ATTIKO General University Hospital, National and Kapodistrian University of Athens, Rimini-1, Haidari, 12462 Athens, Greece

**Keywords:** graft-versus-host disease, allogeneic stem cell transplantation, endothelial cell, dysfunction, biomarkers

## Abstract

**Simple Summary:**

Hemato-poietic stem cell transplantation is associated with significant endothelial dysfunction, which may result in severe complications. Endothelial dysfunction induces the expression of multiple substances including, cell adhesion molecules, pro-inflammatory cytokines, and coagulation factors resulting in activation of the complement system, and promoting a procoagulant and proinflammatory state. The pathophysiological process that mediates the endothelial damage is not clearly understood and the aim of this study is a comprehensive review of the existing knowledge. Moreover, the levels of soluble molecules released in the systemic circulation as a result of endothelial damage may serve as biomarkers for the early diagnosis and the pre-emptive treatment of post-transplant syndromes. The review of the existing literature presented in detail shows that the ideal biomarker with sufficient sensitivity and specificity for the early diagnosis of post-transplant complications does not currently exist. Future studies should focus on the identification of novel biomarkers with sufficient specificity and sensitivity.

**Abstract:**

Allogeneic hematopoietic stem cell transplantation (allo-HSCT) remains the only therapy with a curative potential for a variety of malignant and non-malignant diseases. The major limitation of the procedure is the significant morbidity and mortality mainly associated with the development of graft versus host disease (GVHD) as well as with a series of complications related to endothelial injury, such as sinusoidal obstruction syndrome/veno-occlusive disease (SOS/VOD), transplant-associated thrombotic microangiopathy (TA-TMA), etc. Endothelial cells (ECs) are key players in the maintenance of vascular homeostasis and during allo-HSCT are confronted by multiple challenges, such as the toxicity from conditioning, the administration of calcineurin inhibitors, the immunosuppression associated infections, and the donor alloreactivity against host tissues. The early diagnosis of endothelial dysfunction syndromes is of paramount importance for the development of effective prophylactic and therapeutic strategies. There is an urgent need for the better understanding of the pathogenetic mechanisms as well as for the identification of novel biomarkers for the early diagnosis of endothelial damage. This review summarizes the current knowledge on the biology of the endothelial dysfunction syndromes after allo-HSCT, along with the respective therapeutic approaches, and discusses the strengths and weaknesses of possible biomarkers of endothelial damage and dysfunction.

## 1. Introduction

Allogeneic hematopoietic stem cell transplantation (allo-HSCT) remains the only curative treatment for many malignant and non-malignant diseases [1]. However, its success its hampered by the development of a series of complications that are related to endothelial injury, such as sinusoidal obstruction syndrome/veno-occlusive disease (SOS/VOD), graft-versus-host disease (GvHD), transplant-associated thrombotic microangiopathy (TA-TMA), etc., resulting in the so-called “endothelial dysfunction (ED) syndromes” [1,2,3]. These syndromes are associated with high morbidity and mortality, making their prompt recognition and treatment of particular importance for improved outcomes.

Endothelial cells (ECs) play an essential role in maintaining vascular homeostasis, controlling vascular tone and permeability, adjusting responses to stress/inflammation, and in regulating coagulation. During allo-HSCT, ECs are confronted by multiple challenges, such as the conditioning chemo-/radiotherapy, the immunosuppressive prophylaxis, the neutropenia-related infections, the alloreactive immune responses, etc., resulting in endothelial dysfunction, which is associated with an increased pro-inflammatory, pro-apoptotic, and pro-thrombotic state. It is thus essential not only to better understand the pathophysiology of endothelial dysfunction syndromes but to also identify markers of ED and target endothelium for developing strategies to prevent and treat post-transplant complications in order to improve outcomes after allo-HSCT.

This review summarizes the current knowledge on the biology of the endothelial dysfunction syndromes after allo-HSCT, along with the respective therapeutic approaches, and discusses the strengths and weaknesses of possible biomarkers of endothelial damage and dysfunction in the context of allo-HSCT.

## 2. Pathophysiology of Endothelial Dysfunction in allo-HSCT

The endothelium is a very active organ formed by a thin layer of heterogeneous cells that delignate the barrier between the circulating blood and other tissues [3]. Normal function of the endothelium is of paramount clinical significance, as it plays a key role in maintaining vascular homeostasis and a balanced coagulation but also in host defense, inflammation, and angiogenesis [1,2,4].

ECs participate in the balance between coagulation and fibrinolysis by producing tissue factor pathway inhibitors. However, when activated, ECs release tissue factor (TF) and von Willebrand factor (vWF), switching the balance towards inhibition of fibrinolysis, which leads to a thrombotic predisposition [1]. This hypercoagulable state has been demonstrated both after allo-HSCT and in association with the endothelial dysfunction syndromes in allo-HSCT [5,6]. 

ECs also produce vasoactive regulatory molecules such as nitric oxide (NO), endothelin, and prostacyclin that adjust vascular tone and vascular permeability, regulating blood fluidity [7]. NO also plays a role in inhibiting inflammation by downregulating the production of proinflammatory cytokines through NF-kβ inhibition [8]. The contribution of NO in the pathogenesis of the endothelial dysfunction systems has been proposed by several studies that have correlated the reduced NO levels with reduced overall survival (OS) and ineffective hematopoietic reconstitution after allo-HSCT [9]. 

Angiopoietin-1 (Ang-1) and -2 (Ang-2) are another category of endothelial growth factors with antagonistic effects that control angiogenesis and vascular homeostasis through interactions with the tyrosine kinase receptor Tie-2, expressed in endothelium [10]. They also participate in the endothelial responses to pro-inflammatory and anti-inflammatory conditions by regulating the permeability of vascular endothelium, the leukocyte adherence to EC, and the expression of anti-apoptotic molecules and other inflammatory mediators. This interplay between endothelium and leukocytes is controlled by molecules such as ICAM-1, VCAM, Selectins, and PECAM that regulate overall the recruitment of immune cells to the sites of inflammation [3,11]. The endothelial response to stress/inflammation involves the activation of ECs, which may be either rapid and transient (type 1) or slower and more sustained (type 2), with a resulting secretion of variable cytokines such as tumor necrosis factor-alpha (TNF-a) and interleukin-1β (IL-1β) [1,4]. This is of importance, as tissue-specific ECs may express specific factors (such as WNT2, VEGF, and matrix metalloprotease 14) essential for tissue repair and homeostasis in response to stress stimuli [12,13]. On the other hand, the inflammatory response of activated ECs may lead to increased leukocyte adherence and vascular edema and to a pro-thrombotic state with detrimental results [1,3].

## 3. Endothelial Dysfunction-Related Post-Allogeneic Stem Cell Transplantation

### 3.1. Transplant-Associated Thrombotic Microangiopathy (TA-TMA)

TA-TMA constitutes one of the most severe complications of allogenic hematopoietic stem cell transplantation (allo-HSCT) and is associated with significant morbidity and mortality [14,15,16]. It is a heterogenous disease, which is characterized by aberrant complement activation, endothelial dysregulation, and microvascular hemolytic anemia [17]. Recently, a three-hit theory was proposed regarding the pathophysiology of the disease [1], which refers to: (1) endothelial vulnerability to damage and complement activation (hit 1), (2) a toxic event (such as the conditioning therapy) injuring the endothelium and initiating the complement cascade (hit 2), and (3) additional insults (such as infection, graft-versus-host disease (GvHD), etc.) exacerbating the complement activation and leading to (widespread) microthrombi formation (hit 3). It may affect any organ [18,19] but primarily the kidneys (proteinuria, hypertension, renal failure) [18,20]; the central nervous system (headache, seizures, confusion, posterior reversible encephalopathy) [21,22]; the gastrointestinal tract (abdominal pain, diarrhea, bleeding) [23,24]; and the lungs (pulmonary hypertension) [25,26]. As a result, the clinical picture is quite heterogenous and of variable severity. Multi-organ disease (MOD), requiring intensive care monitoring and treatment, belongs to the very severe end of the spectrum and is associated with very high morbidity and mortality [18]. Overall, prognosis is poor, with case fatality varying between 50–75% [16,27,28].

The precise incidence of the disease remains unknown and varies significantly (from 0.5% to 78%), which is partly due to the lack of consensus in diagnostic criteria [21]. In the last two decades, several groups such as the Blood and Marrow Transplant Clinical Trials Network (BMT-CTN) [29] and the International Working Group (IWG) [30] have attempted to define specific diagnostic criteria for TA-TMA, but they were both found to carry significant weaknesses and limitations in a subsequent validation study by Cho et al. [31]. Another study by Shayani et al. [32] divided the patients into those with “possible” and “probable” TA-TMA based on different (the “City of Hope”) criteria, whereas Postalcioglu et al. [15], concluded that clinical TA-TMA was significantly under-recognized using these criteria, translating to poor outcomes due to the lack of prompt therapeutic intervention. Finally, Jodele et al. [19] attempted to define “high-risk” disease in a prospective pediatric study and concluded that proteinuria (>30 mg/dL) and elevated serum C5b-9 (as a marker of complement activation) were able to predict poor outcomes (84% NRM at 1 year). The criteria proposed by different study groups for the diagnosis of TA-TMA are shown in Table 1.

Given the significant morbidity and mortality associated with TA-TMA, the identification of risk factors associated with the development of the disease had been the focus of extensive investigation since the initial description of TA-TMA as a side effect of cyclosporin in 1980 [33]. These have been roughly categorized in pre- and post-transplant risk factors as follows: (i) pre-transplant risk factors: age, mismatched donor, myeloablative conditioning, non-malignant disorders, and history of prior allo-HSCT and (ii) post-transplant risk factors: acute GvHD, (x4-fold increase), high disease risk index, high baseline LDH, elevated CNI levels, infections (CMV, invasive aspergillosis, BK viremia, bacteremia), mammalian target of rapamycin inhibitors (mTORi), and venous thromboembolic disease [18,34]. Since most of these factors cannot be modified or avoided, a number of medications has been explored in an effort to prevent the development or minimize the severity of TA-TMA. Prophylactic ursodeoxycholic acid (UDA) was found to reduce non-relapse mortality (NRM) and severe aGvHD in a prospective randomized study, whereas statin prophylaxis, with or without the concomitant use of UDA, was proven to be safe and reduce the risk of TA-TMA, improving outcomes [1,35]. This has led several transplant centers to adopt a statin-based prophylactic approach combining UDA and pravastatin [1,36].

With regards to treatment, preventative measures (such as HLA-matched transplantation, avoidance of toxic medications, use of reduced intensity conditioning, etc.) and supportive care are of paramount importance [1,37,38]. Prompt withdrawal of CNIs is also advised as primary intervention for TA-TMA management [29] although in a study by Li et al., this approach failed to improve survival [34]. This was attributed to the exacerbation of the underlying GvHD, which is also associated with increased morbidity, underlying the need for careful consideration of alternative substitution strategies upon CNI withdrawal. The value of therapeutic plasma exchange (TPE), once the gold standard, is debatable, as low success rates (~6%) have been documented by two different studies [39,40]. Defibrotide (DF) has been associated with favorable outcomes, showing promise in the treatment of TA-TMA [41,42,43]. More interestingly, Higham et al. used DF as prophylaxis, with significantly reduced incidence of TA-TMA (4%) compared to historical values (18–40%) [44]. The pivotal role of complement activation in the pathophysiology of TA-TMA has led complement inhibitors to the top of the treatment ladder. The use of Eculizumab has been associated with significantly better response rates and overall survival, constituting it a first-line agent in many institutes [45,46]. However, it is of particular importance that Eculizumab trough and CH50 levels are closely monitored, as more intensive treatment (dose and frequency) is required for successful outcomes [47,48]. Narsoplimab, a MASP-2 inhibitor, received a breakthrough designation from the FDA in 2021 for the treatment of TA-TMA in view of the significant improvements in hematologic parameters and overall survival [18].

### 3.2. Sinusoidal Obstructive Syndrome (SOS)/Veno-Occlusive Disease (VOD)

Sinusoidal obstructive syndrome (SOS), also known as veno-occlusive disease (VOD), is a life-threatening complication occurring after high-dose chemotherapy and HSCT [49,50]. It has also been described after high-dose radiotherapy, liver transplantation, and administration of toxic agents [51,52,53]. In HSCT, the conditioning regimen causes an initial toxic injury to the sinusoidal endothelium of the liver, disrupting the endothelial cohesions and creating gaps in the sinusoidal barrier. This allows red blood cells, leukocytes, and other debris to pass through and accumulate into the Disse space, leading to dissection of the endothelial lining and downstream embolization and sinusoid flow obstruction. This results in reduced hepatic outflow and post-sinusoidal hypertension with subsequent hepato-renal syndrome and MOD [54,55].

Clinically, SOS/VOD is characterized by weight gain (unresponsive to diuretics), painful hepatomegaly, ascites, and jaundice (although anicteric cases have been reported) [56]. Its severity varies from mild/self-resolving to severe (~25–30% patients), with MOD involving the kidneys (hepatorenal syndrome), the lungs (hypoxia, pleural effusion, pulmonary infiltrates), and the central nervous system (encepathopathy) [57]. MOD is associated with very high mortality rates (up to 80%), imperatively constituting the need to identify predictive factors for severe disease.

Several risk factors for SOS/VOD have been identified in recent years and are broadly divided into patient- and transplant-related as follows: (i) patient-related: older age, Karnofsky score <90%, pre-existing liver disease, impaired liver function tests (transaminases >×2 upper limit of normal (ULN) and bilirubin >×1.5 ULN, advanced disease, thalassemia, prior transplant, abdominal or hepatic radiation, metabolic syndrome, and raised ferritin and (ii) transplant-related: allogenic transplant, unrelated donor, HLA-mismatched donor, T-cell replete transplant, and myeloablative conditioning (containing either busulfan or total body irradiation) [58,59]. The use of novel immunotherapies for the treatment of acute leukemias, such as gemtuzumab ozogamicin and inotuzumab ozogamicin, have also been linked to the development of SOS/VOD, necessitating particular vigilance when using those agents pre-transplant [60,61].

The incidence of SOS/VOD after allo-HSCT varies significantly (from 5% to 67%) owing to different patient cohorts and transplant procedures and to different diagnostic criteria applied in different centers [57,62]. It usually develops within the first 21 days after allo-HSCT although in 15–20% patients, it may occur later [63]. Historically, the diagnosis of SOS/VOD was based on the appliance of either of the Baltimore or the modified Seattle criteria [64,65]. Although both require that patients must be within 3 weeks post transplant and include common manifestations of the disease, the main difference between the two is the inclusion of hyperbilirubinemia, which is included but not required in the Seattle criteria. Several studies have supported the use of modified Seattle criteria versus the Baltimore criteria in regard to SOS/VOD prediction, implying that waiting for hyperbilirubinemia to develop may allow progression to more severe disease, leading to worse outcomes [66,67]. In view of the aforementioned conflicting definitions leading to delayed diagnosis with significant impact on outcomes, along with the increased frequency of late-onset SOS/VOD due to the reduced intensity conditioning and the alternative donors used, the EBMT consortium established the updated EBMT criteria for SOS/VOD diagnosis in pediatric and adult populations [68,69]. The EBMT diagnostic criteria include an early and a late-onset SOS/VOD, with histology and imaging (ultrasound) having a key role in establishing the diagnosis itself. They also proposed criteria for severity grading (mild, moderate, severe, and very severe) once the diagnosis is made. This was validated in a subsequent study of 203 patients confirming significantly higher TRM in very severe SOS/VOD although further validation may be required [70]. 

With regards to the treatment of SOS/VOD, supportive care with close clinical monitoring (as rapidly developing disease) and timely initiation of defibrotide (DF) therapy are of paramount importance. Supportive care includes daily reports of weight, urinary output, abdominal circumference, etc., as well as therapeutic measures to comfort the patient (such as diuresis, paracentesis or thoracentesis, oxygen and analgesic therapy, etc.) [59]. Defibrotide, an oligonucleotide with anti-thrombotic, anti-inflammatory, and anti-ischemic activity, is the only approved drug for the treatment of SOS/VOD [71]. The dose of 25 mg/kg/day was evaluated in a multicenter phase III study, which showed significantly higher CR rates (24% vs. 9%) and day +100 OS rates (38% vs. 25%) in the treatment group, without any differences in the side-effect profile [68]. This dose was further validated in a multicenter prospective study confirming that the use of 25 mg/kg/day, for at least 21 days and until resolution of symptoms, was associated with the best outcomes with the least toxicity [72]. With regards to timing of DF treatment, several studies have proposed that prompt initiation of DF is associated with better outcomes [73,74]. It is therefore recommended that patients with moderate SOS/VOD should be considered for DF treatment, whereas patients with mild disease should be closely monitored in case of deterioration. 

In terms of prophylaxis, ursodeoxycholic acid (UDA) has been associated with a significant reduction of SOS/VOD incidence in various randomized studies [75,76,77]. The use of DF as a prophylactic agent has also been shown to reduce the incidence of VOD/SOS in high-risk patients by several studies [78], whereas a systematic review by Zhang et al. confirmed the lower relative risk of SOS/VOD with DF prophylaxis (risk ratio 0.47, 95% CI) [79]. A more recent meta-analysis by Cobacioglu et al. confirmed a low incidence of SOS/VOD following IV DF prophylaxis regardless of age group, supporting the use of DF in the prevention of SOS/VOD [78]. However, a prospective phase III study of DF prophylaxis for SOS/VOD (NCT02851407) stopped enrolment after meeting the criteria for futility although analyses are ongoing and are awaited with great interest.

### 3.3. Lung Injury Syndromes

Idiopathic pneumonia syndrome (IPS) was defined in 1993 in a workshop organized by the National Institute of Health as the result of widespread alveoli injury with multi-lobar pulmonary infiltrates and symptoms related to respiratory failure. 

IPS may present with a variety of clinical symptoms depending on the site of lung injury. However, the typical presentation is that of acute interstitial pneumonitis. Other manifestations include diffuse alveolar hemorrhage (DAH) and peri-engraftment respiratory distress syndrome (PERDS) [80,81,82,83,84]. Table 2 shows the clinical features of lung injury syndromes, while the criteria used for the differential diagnosis between these syndromes are presented in Figure 1.

#### 3.3.1. Idiopathic Pneumonia Syndrome

IPS, with the most typical manifestation of acute interstitial pneumonitis, occurs during the first weeks after stem cell transplantation. The incidence ranges from 3% to 15% after allo-HSCT with the use of myeloablative conditioning [85,86]. However, it is significantly lower after reduced intensity conditioning as well as after autologous HSCT [87]. The median time of onset is 30–40 days (range, 14–100) after allo-SCT although more recent studies have suggested an even earlier time of onset [88]. The prognosis is extremely poor, with mortality rates of approximately 80%, while the death rate is close to 100% for those patients who require mechanical ventilation [89]. IPS after autologous HSCT is a different entity with probably different pathogenesis, which tends to occur later post transplant and has a favorable response to corticosteroids [90,91]. 

Previous studies have proposed the following risk factors to be associated with the development of IPS after allo-HSCT: (1) intensity of the conditioning, (2) total body irradiation as part of the conditioning, (3) presence of acute GVHD, (4) older age of the recipient, and (5) diagnosis of acute myeloid leukemia or myelodysplastic syndrome [92,93].

Although the pathogenesis of IPS after allo-HSCT remains elusive, it is thought to be the result of microvascular endothelial and alveolar epithelial cell injury, which is mediated by the toxicity of the conditioning regimen in combination with the immune damage induced by alloreactive donor T cells [94,95,96,97].

#### 3.3.2. Diffuse Alveolar Hemorrhage

Diffuse alveolar hemorrhage (DAH) is a distinct subtype of IPS characterized by progressive accumulation of red blood cells in BAL samples and/or more than 20% of hemosiderin laden alveolar macrophages in at least 30% of the alveolar spaces [98]. Clinically, DAH is characterized by rapid respiratory deterioration unless adequately treated [99]. However, the presence of alveolar hemorrhage is not synonymous to the DAH syndrome since it may be observed in the context of lung injury due to various causes, such as lung infections. It is therefore necessary to perform a complete and comprehensive work up for the presence of an occult infection prior to the establishment of diagnosis [100]. Although the median time of DAH onset is 20–30 days post graft infusion, it usually develops at a median of two weeks post engraftment, suggesting that the neutrophils are key players in the disease pathogenesis [101]. The incidence of DAH is equal between allo-SCT and auto-SCT and occurs in approximately 5–10% of patients [102]. Risk factors associated with the development of DAH after transplantation are: (1) the graft source (DAH is observed more frequently with the use of cord blood); (2) the intensity of the conditioning, with myeloablative conditioning and/or the administration of TBI being the most important high-risk factors; (3) the older age of the recipient; and (4) the occurrence of primary graft failure or the delayed engraftment of neutrophils and/or platelets [99,100,103]. Recently, a DAH case in a healthy stem cell donor possibly associated with granulocyte colony stimulating factor (G-CSF) administration was reported and raised concerns regarding the contribution of G-CSF in the pathogenetic process [104].

With regards to pathogenesis, it has been suggested that the syndrome is the result of an initial alveolar injury mediated by the toxicity of the conditioning regimen that is further aggravated during engraftment by the inflammatory potential of neutrophils and monocytes infiltrating the lung. Increased levels of various cytokines such as IL-12, G-CSF, and TNF-α and of lipopolysaccharides in BAL samples have been associated with the development of DAH in patients after transplantation [105,106]. However, due to the absence of large studies, the exact pathogenesis of DAH in the setting of stem cell transplantation remains largely unknown. 

No specific treatment is available for DAH, and its therapy is largely based on the empirical use of high-dose corticosteroids [107] owing to the suspected role of inflammation in the pathogenetic process. However, the efficacy of steroids is less than modest, and the exact dose is still a matter of debate [108]. Supportive care early in the course of the disease is of paramount importance and includes platelet transfusions and hemostatic factors such as aminocaproic acid and recombinant factor VIIa. Despite all efforts, the overall mortality rate remains high and ranges from 60% to 100% [109,110,111].

#### 3.3.3. Peri-Engraftment Respiratory Distress Syndrome

Peri-engraftment respiratory distress syndrome (PERDS) is a manifestation of acute lung injury that occurs in the setting of hematopoietic stem cell transplantation [112]. It is a severe form of engraftment syndrome (ES), which is a systemic capillary leak syndrome occurring at the peri-engraftment period, defined as the period within 3 days before and 7 days after neutrophil reconstitution. PERDS is characterized by hypoxemia and bilateral pulmonary infiltrates as a result of non-cardiogenic pulmonary edema [113,114]. The incidence of PERDS varies widely from as low as 2% to as high as 20% due to differences in the diagnostic criteria but also mostly due to the various patient populations included in the existing clinical studies [113,114,115].

Risk factors associated with the development of PERDS are: (1) female sex, (2) the source of the graft (most common after PBSC than bone marrow grafts), (3) the intensity of pre-transplant chemotherapy (the less the intensity, the higher the probability of developing PERDS), and (4) the use of GM-CSF for accelerating engraftment [116,117,118]. PERDS is observed more frequently after auto-HSCT than allo-HSCT and more often after auto-HSCT for autoimmune diseases. [119]. 

PERDS usually presents in the context of ES and is commonly associated with systemic inflammatory clinical and laboratory signs such as fever, skin rash, weight gain, due to fluid retention, and elevated C-reactive protein. The exact pathogenetic mechanism of PERDS has not been fully elucidated. However, existing data support the role of activated myeloid cells, including neutrophils and monocytes, as the major players in the pathogenetic process. Activated myeloid cells infiltrate the lung and secrete a cocktail of proinflammatory cytokines including IL-1β, IL-2, and IL-6 that induce endothelial cell damage in the lung microvasculature. [120]. The administration of GM-CSF after graft infusion is associated with faster recovery of activated myeloid and dendritic cells that contribute to the induction of capillary leak syndrome [121]. 

PERDS is generally associated with a favorable prognosis. It has an excellent response to corticosteroids although many mild cases tend to resolve spontaneously. Data from previous studies showed increased incidence of acute GVHD and increased mortality in the first year after transplant in patients with a previous diagnosis of ES/PERDS [122].

## 4. Vascular Endothelium: A Target of Graft-Versus-Host Disease

Traditionally, GvHD has been considered as an epithelial cell disease, with the three most commonly involved organs being the skin, intestinal tract, and liver. However, experimental and clinical data allow us to reconsider the epithelium as being the single GVH target.

Indeed, host vascular ECs is the first cell type that the alloreactive donor T lymphocytes attack after allo-HSCT, potentially inducing endothelial GvHD. In accordance with the previous notion, many clinical syndromes observed after allo-HSCT are pathogenetically considered as syndromes of endothelial cell dysfunction.

The term “endothelial cell dysfunction” is poorly defined and is used for the description of certain changes in endothelial cells (EC). There are several stimuli that have the potential to induce EC dysfunction especially in the setting of allo-SCT, such as infections, toxins, chemotherapeutic agents, donor cell alloreactivity, etc. [123]. Intensive and persistence activating stimuli can produce either organ-localized or systemic EC dysfunction. EC dysfunction in the setting of allo-HSCT can present with the form of specific endothelial syndromes such as transplant associated microangiopathy, engraftment syndrome, diffuse alveolar hemorrhage, etc. EC dysfunction results in distortion of normal homeostasis, increase in capillary permeability, and extravasation of fluid, plasma proteins, and immune cells from the vascular space to extracellular tissue, thus creating an inflammatory milieu.

The pathophysiology is not clearly delineated and may involve interactions of EC with T cells, monocytes, complement components, and proinflammatory cytokines [124]. 

Experimental data have shown that EC are immune targets of alloreactive donor T cells, and EC dysfunction syndromes observed after allo-HSCT can be viewed as “graft-versus-EC” phenomena [125].

Moreover, previous studies in humans have shown that host ECs are a target of alloreactive donor cytotoxic T lymphocytes. Patients with GVHD had an extensive loss of microvessel density in skin biopsies as compared to patients without GVHD and normal donors. Moreover, patients with GVHD had histological signs of vascular injury such as endothelial swelling or denudation. As a further proof of the concept, researchers observed the presence of activated CD8 cytotoxic cells in close vicinity to damaged endothelium. Extensive microvessel loss results in impaired blood perfusion and tissue damage and is possibly associated with the development of tissue fibrosis [126].

Moreover, in inflammatory conditions, activated ECs acquire the potential to function as antigen-presenting cells and therefore contribute to the initial stimulation and expansion of alloreactive donor T cells, thus further aggravating the graft-versus-host process [127].

## 5. Atheromatosis: A Late GVHD Manifestation

Long-term survivors after allo-HSCT have an increased incidence of cardiovascular events when compared with healthy controls. An obvious explanation is the presence of metabolic syndrome occurring frequently in patients treated with HCT. Allogeneic HSCT survivors were more likely to report hypertension, diabetes, and dyslipidemia than patients after autologous HSCT [128]. The increased occurrence of dyslipidemia, hypertension, and diabetes after allo-HSCT can be attributed to prolonged administration of calcineurin inhibitors and corticosteroids or can be the result of subclinical organ damage such as hypothyroidism or decreased growth factor hormone [129]. A previous study including more than two hundred long-term survivors after allo-HSCT showed that the cumulative incidence of cerebrovascular disease was extremely high and, more importantly, occurred at a younger median age than the usual. When adjusted for age, patients after allo-HSCT had a significantly increased risk of arterial events, while the risk for cerebrovascular disease of long-term survivors after autologous HSCT was not different from the age-matched general population [130]. Based on these data, long-term survivors should be carefully followed by experienced specialists for the early diagnosis and treatment of metabolic diseases such as diabetes, dyslipidemia, etc. 

Atherosclerosis pathogenetically is the result of an inflammatory process that affects the endothelium. Atheromatous lesions occur gradually during the decades before the clinical manifestations of stroke, peripheral arteriopathy, and coronary heart disease. Based on the above data, it is highly possible that the premature atheromatosis observed after allo-HSCT is the final result of the graft-versus-host effect and the immune attack on host epithelium. The late occurrence is explained by the long latency between the initial stimuli and the development of arterial events. Premature atheromatosis after allo-SCT can be considered as a manifestation of the graft-versus-endothelium process.

A similar allo-immune effect occurs after solid organ transplantation, where the most important cause of chronic organ failure is chronic vasculopathy that involves the arteries of the transplanted organ [131,132]. Although chronic transplant vasculopathy is considered a multifactorial disease, immune-mediated endothelial damage seems to be the most important factor. In the setting of solid organ transplantation, vascular endothelial cells are one of the major targets of cellular and humoral alloreactivity [133].

Although supported by observational studies and multiple case reports, the direct association between GVHD and premature atheromatosis has not been definitely established. A schematic overview of the endothelial dysfunction-associated syndromes after allo-HSCT is given in Figure 2.

## 6. Soluble Biomarkers of Endothelial Dysfunction

Endothelial damage syndromes occurring after allo-HSCT usually overlap with other complications, and therefore, biomarkers are needed for early identification of endothelial-cell injury. Endothelial dysfunction results in loss of vascular integrity and increase in capillary permeability. EC damage induces changes in various protein expression, shedding of these proteins from damaged endothelial cells, and release in the systemic circulation. Monitoring of released proteins in the peripheral blood may help in the early identification of endothelial cell damage. 

The ideal biomarker for EC dysfunction should be characterized by high sensitivity, specificity, and predictive value. Although an ideal biomarker does not currently exist, there is an enormous activity in the field, with various soluble molecules, circulating endothelial cells, coagulation factors, cytokines, etc., being actively tested. 

### 6.1. Von Willebrand Factor

Von Willebrand factor (VWF) is a protein that plays an important role in microvascular integrity and hemostasis. VWF is a bridging molecule for platelet adherence to damaged endothelium as well as for platelet aggregation and the formation of platelet thrombi at the site of vascular injury. Moreover, VWF is the carrier for FVIII and helps in the transportation in sites of injury and the local production of thrombin. VWF is produced in endothelial cells and megakaryocytes. VWF is stored in granules of platelets, while large quantities of VWF are stored in Weibel–Palade bodies in endothelial cells. Endothelial dysfunction induced by various stimuli results in secretion of VWF from Weibel–Palade bodies to systemic circulation [134].

Elevated plasma levels of VWF have been reported in various inflammatory conditions where endothelial dysfunction has a prominent role, such as autoimmune and neoplastic disorders. As an example, VWF is a predictor for pulmonary involvement in patients with systemic sclerosis, while the levels of VWF correlate with disease activity in connective tissue disorders and hemolytic/uremic syndrome [135,136,137].

Plasma levels of VWF are significantly increased early after stem cell transplantation, possibly as a result of chemoradiotherapy-induced endothelial damage [138]. As a further proof of the concept that endothelial damage is a major contributor to chronic GVHD pathogenesis, previous studies have shown that VWF is observed during chronic GVHD episodes, and more importantly, VWF levels correlate with disease activity and severity [139,140].

Moreover, few studies have shown that elevated levels of VWF were predictive for the development and severity of endothelial dysfunction syndromes observed after allo-HSCT, such as transplant-associated thrombotic microangiopathy (TA-TMA) [141].

Plasma levels of VWF correlate with the degree of endothelial damage factor (vWF). In a previous study including 79 consecutive patients who underwent allo-HSCT, levels of VWF were prospectively monitored. TA-TMA developed in 23 (29%) out of 79 patients, and these patients had significantly higher non-relapse mortality (NRM) as compared to patients without TA-TMA. Plasma levels of VWF were significantly higher in patients with TA-TMA, and ROC curve analysis showed the predictive ability of VWF for the diagnosis of TA-TMA. In multivariate analysis, patients with elevated VWF had a higher risk of developing TA-TMA and a decreased overall survival. Authors concluded that VWF monitoring is a useful predictor of TA-TMA and can help in the early diagnosis [142].

Elevated levels of VWF and FVIII have been observed in patients with sinusoidal obstruction syndrome (SOS) after allo-SCT. However, the prognostic utility of these markers in SOS has not been established [69].

### 6.2. Cellular Adhesion Molecules

Cellular adhesion molecules (CAMs) such as ICAM, VCAM, E-Selectin, and P-Selectin are expressed on endothelial cells and exist in two isoforms, namely the membrane-bound and soluble form, and are important mediators for leucocyte adhesion to endothelium and transmigration from blood to tissues [143]. 

Previous studies examined the levels of soluble forms of VCAM, ICAM, and E-selectin in patients after allo-SCT. Patients with acute GVHD had significantly higher serum levels of s-ICAM and E-selectin, while levels of s-VCAM and E-selectin were significantly increased in chronic GVHD as compared with patients without GVHD. Elevations of serum levels preceded the onset of clinical symptoms by at least 30 days in most cases. Similarly, s-VCAM and E-selectin were also significantly increased in patients with TAM [144]. 

### 6.3. Thrombomodulin and Plasminogen Activation Inhibitor-1

Thrombomodulin (TM) is a high-affinity thrombin receptor expressed on the endothelial cell surface and is an important natural anticoagulant. It acts as a cofactor of activated protein C, and its major role is the inhibition of procoagulant functions of thrombin. Plasminogen activation inhibitor-1 (PAI-1) is a serine protease inhibitor that functions as the principal inhibitor of tissue plasminogen activator (tPA) and urokinase (uPA). tPA and uPA convert plasminogen to plasmin, and therefore, PAI-1 functions as an inhibitor of fibrinolysis. PAI-1 is mainly produced by endothelial cells but is also produced by other cells such as the cells of adipose tissue.

Serum levels of TM in combination with VWF, s-ICAM1, and E-selectin were significantly increased in patients who developed SOS or TAM after allo-SCT. In more detail, serum levels of these soluble molecules measured one week after graft infusion were predictive of subsequent development of SOS [145,146,147].

Similarly, serum levels of PAI-1 were significantly increased in patients with SOS but not in patients with GVHD. Moreover PAI-1 is also a predictor for the outcome of SOS after treatment with defibrotide (DF). A previous study showed that the decrease in levels of PAI-1 in the first two weeks after DF initiation was correlated with the subsequent complete remission of SOS. PAI-1 is a useful marker for the diagnosis of SOS and treatment outcome after DF but is also useful for the differential diagnosis of SOS from liver acute GVHD [146,148,149].

### 6.4. Circulating Angiogenic Factors

Angiopoetin-1 (Ang-1) and Ang-2 are angiogenic peptides that act in a pair as agonist and antagonist and are major regulators of endothelial homeostasis. The peptides are ligands for the tyrosine kinase receptor Tie2 expressed on the endothelial surface. Binding of Ang1 with Tie2 promotes EC survival and permeability, whereas it induces the downregulation of surface adhesion molecules, thus helping in the suppression of inflammation. 

The antagonist peptide Ang2, after interacting with Tie2, induces the opposing effects by promoting EC apoptosis and stimulating the expression of inflammatory cytokines. Many studies have shown that Ang2 is a predictor of systemic endothelial damage and the associated transplant complications [150].

The incidence of transplant-related complications associated with endothelial cell damage, such as SOS, TAM, IPS, and DAH, have been correlated with serum levels of Ang2. Increased Ang2 levels at the time of transplant were associated with the late development of endothelial damage syndromes in a group of 153 allo-SCT recipients. Multivariate analysis showed that high levels of Ang2 were also predictive of increased EC-syndrome-associated mortality [150,151].

On the contrary, serum levels of Ang2 were not predictive for the development of acute GVHD. However, increased levels of Ang2 correlated with refractoriness to treatment of severe acute GVHD as well as with increased GVHD-associated mortality [152]. 

Regarding vascular endothelial growth factor (VEGF), previous studies have shown that allo-HSCT recipients tend to have lower VEGF serum levels as compared with healthy donors. However, in the same studies, it was observed that VEGF levels are significantly increased in patients with GVHD. These studies showed that levels of VEGF are elevated at the time of GVHD but not before clinical diagnosis, and therefore, VEGF is not considered a predictive marker for GVHD [153,154].

### 6.5. Proinflammatory Cytokines

Pro-inflammatory cytokines such as IL2, IL6, IL33, IFNg, and TNFa are important mediators of the inflammation and play a significant role in the generation and enhancement of alloreactive reaction resulting in EC damage and the development of GVHD [155,156].

Previous studies have shown that increased serum levels of TNFa, soluble IL2 receptor-a (Sil2Ra), IL6, and IL33 have been observed during the course of GVHD. However, increased levels of these cytokines are also observed in other settings of inflammation such as sepsis and also in other transplant-associated complications such as SOS [157,158,159]. Therefore none of these molecules can serve as a biomarker for GVHD [160,161].

### 6.6. GVHD Biomarkers Discovered by Proteomics

The Mount Sinai Acute GVHD International Consortium (MAGIC), after performing proteomic analysis in hundreds of patients with and without GVHD after allo-HSCT, discovered biomarkers that help in the diagnosis and prediction of acute GVHD. Among them, two serum biomarkers, i.e., the regenerating islet-derived 3a (REG3a) and suppressor of tumorigenesis 2 (ST2), are released in the systemic circulation from Paneth and intestinal crypt cells damaged during the GVHD process. Based on the serum levels of these two biomarkers, they developed the MAGIC algorithm probability (MAP) and validated its utility as a diagnostic and predictive for a GVHD outcome biomarker. Researchers showed that MAP is actually the best predictor of the outcome of acute GVHD (better than overall grade) and has the potential to guide therapy [162].

### 6.7. Endothelial Activation and Stress Index (EASIX)

An easy, reproductive, and useful tool for predicting outcome after allo-SCT is the endothelial activation and stress index (EASIX). EASIX is a new biomarker panel based on the estimation of three easily available parameters, i.e., creatinine, platelet count, and LDH. EASIX score is calculated using the formula [(Lactate Dehydrogenase (U/L)) x (Creatinine (mg/dL))/Thrombocytes (10^9^ cells/L)] [36,163]. Previous studies have shown that elevated EASIX score at the time of graft infusion is predictive for the risk of death in patients with acute GVHD and TAM [164,165]. Furthermore, EASIX is a good predictive biomarker for the development of SOS [166].

## 7. Cellular Biomarkers of Endothelial Dysfunction

Emerging evidence suggest that cellular-based markers such as circulating endothelial cells (CECs), endothelial progenitor cells (EPCs), and extracellular vesicles (EVs) represent a novel approach to assessing endothelial function. 

CECs are mature endothelial cells that are released from the intima after EC injury and loss of EC integrity. In healthy individuals, they constitute rare events (~5 cells/μl), but their plasma levels increase significantly in the context of vascular injury, reflecting the degree of endothelial damage [167,168]. They are characterized by the expression of endothelial markers (such as CD31, CD144, CD146, and vWF) as well as the presence of VEGFR2 and CD133 and the absence of leucocyte markers. However, their phenotype as well as their morphology may vary depending on the type of the underlying disorder. In the context of allo-HSCT, a significant increase in CEC numbers compared to baseline has been associated with the administration and type of the conditioning [169,170]. More specifically, total body irradiation (TBI) resulted in a much higher rise of CECs (median 44 cells/μl) compared to RIC (median 24 cells/μl), which is in line with the concept of radiation-induced endothelial damage [171]. Interestingly, the CEC levels were higher than the baseline even before the conditioning therapy, suggesting either a tumor-related or a chemotherapy-induced event. Finally, an inter-patient variability in CEC numbers was noted, which was attributed to the individual endothelial vulnerability to damage. Touzot et al. correlated elevated CEC levels with a number of post-transplant complications, such as the development of SOS/VOD, TA-TMA, capillary leak syndrome, and pulmonary artery hypertension, indicating the role of CECs as a biomarker of endothelial damage. The diagnostic potential of CECs was further established by Almici et al., who demonstrated that elevated CEC levels are able to predict the development and therapy response of GvHD [172,173].

In contrast to CECs, which are released in response to injury and are thus considered biomarkers of endothelial damage, EPCs contribute to neo-angiogenesis and vascular repair. They are immature, precursor cells that are derived from the bone marrow and characterized by early lineage (such as CD133, CD34) and common endothelial markers (such as CD117, CD31, VEGFR2, vWF) and their capacity to differentiate into endothelial cells in vitro [174,175]. Low levels of EPCs have been detected in a number of diseases, such as cardiovascular disease, chronic renal failure, hypertension, etc., constituting them a surrogate marker of cardiovascular risk [176]. Similarly, low levels of EPCs have been described in patients undergoing allo-HSCT (even before the transplant), along with raised CECs, reflecting an ongoing endothelial damage. More interestingly, the persistence of low EPC and high CEC levels beyond 12 months of allo-HSCT may explain the higher cardiovascular risk associated with long-term survivors of allo-HSCT although larger studies are required to confirm this association [177].

EVs are a heterogenous population of a-nuclear, membrane-enclosed particles that carry variable bio-acting molecules such as miRNA, mRNA, DNA, proteins, lipids, etc. They have emerged as important mediators of intercellular communication and homeostasis owing to their unique functions reflecting that of the parental cell [178]. Endothelial cells constitutively secrete low-level EVs under physiological conditions, which increase significantly under EC stress and injury [179]. In the context of allo-HSCT, elevated CD144+ EVs have been detected in early stages of SOS/VOD, reflecting early endothelial damage in the disease pathogenesis [180]. However, although these results are very promising, larger prospective studies are required to validate them prior to clinical application. A synopsis of the studied biomarkers for endothelial dysfunction syndromes is presented in Table 3.

## 8. Conclusions

The endothelium is a key player in maintaining normal homeostasis by regulating blood flow, capillary permeability, coagulation, and inflammatory response. Hematopoietic stem cell transplantation is associated with significant endothelial activation, which in many occasions results in severe endothelial dysfunction-related post-transplant complications. Multiple stimuli mediate the endothelial damage, such as the toxicity of conditioning regimen, the use of calcineurin inhibitors, certain infections, and the donor cell alloreactivity against host endothelial cells. Endothelial dysfunction induces the secretion of coagulation factors (VWF and TM) the expression of cell adhesion molecules (ICAM, VCAM) and pro-inflammatory cytokines (TNFa, IFNγ, etc.), while it also activates the complement system. The levels of soluble molecules released in the systemic circulation as a result of endothelial damage may serve as biomarkers for the early diagnosis and the pre-emptive treatment of post-transplant syndromes. However, an ideal biomarker with sufficient sensitivity and specificity for the early diagnosis of post-transplant complications does not currently exist. Future studies should focus on the identification of specific biomarkers for the early diagnosis of endothelial dysfunction-associated post-transplant syndromes. The understanding of the pathophysiological process that mediates the endothelial damage after hematopoietic stem cell transplantation will be of fundamental importance for the design of effective therapeutic interventions for the prevention of transplant-associated complications.

## Figures and Tables

**Figure 1 cancers-15-00680-f001:**
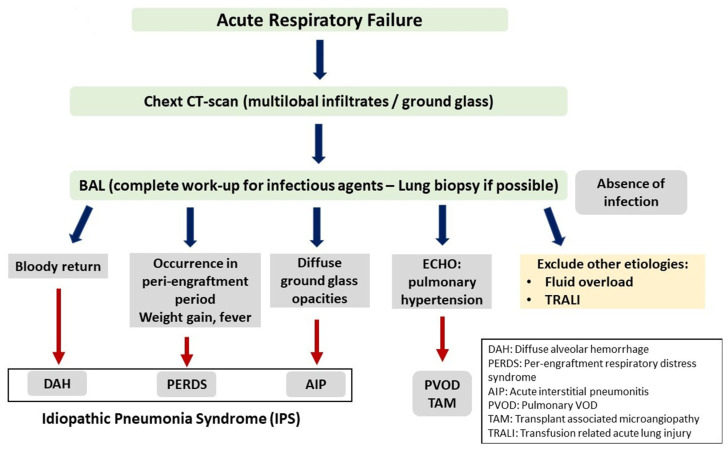
Idipathic Pneumonia Syndrome.

**Figure 2 cancers-15-00680-f002:**
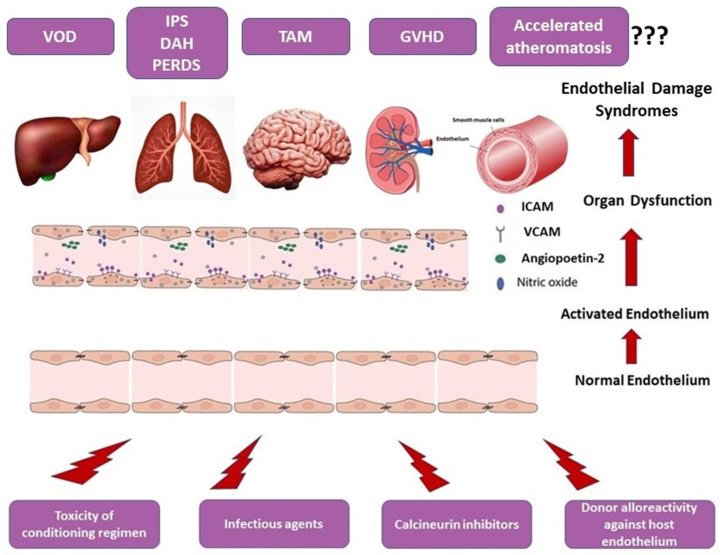
Pathogenesis of endothelial dysfunction syndromes.

**Table 1 cancers-15-00680-t001:** Proposed diagnostic criteria for thrombotic microangiopathy.

Criteria	BMT-CTN, 2005	IWG, 2007	Jodele, 2014
Parameter	All Criteria Present	All Criteria Present	≥4 of 7 Criteria Present in ≥2 Occasions in 14 Days
Anemia or increasing RBC transfusion requirements	YES	YES	YES
New-onset thrombocytopenia, >50% decrease in PLT count, increase in PLT transfusion requirements	YES	YES	YES
Presence of schistocytes	YES (≥2 per HPF)	YES (>4%)	YES
Elevated LDH	YES	YES	YES
Decreased haptoglobin	-----	YES	-----
Hypertension	-----	-----	YES
Proteinuria	-----	-----	YES
Elevated sC5b-9	-----	-----	YES
Renal dysfunction	YES	-----	-----

The presence of end-organ damage is a marker of severity; renal biopsy is not required for diagnosis, but if obtained, it is then sufficient for the establishment of diagnosis.

**Table 2 cancers-15-00680-t002:** Clinical features of acute lung injury syndromes post stem cell transplantation.

Characteristic	Idiopathic Pneumonia Syndrome	Diffuse Alveolar Hemorrhage	Peri-Engraftment Respiratory Distress Syndrome
Epidemiology	More common after allo-SCT	Equal incidence after Auto and allo-SCT	More common after auto-SCT
Median time of onset	30–40 days after allo-SCT	20–25 days after SCT	From 3 days before to 7 days after engraftment
Relation to engraftment	No relation	No relation	Occurs during the peri-engraftment phase
Clinical features	Rapid progression to respiratory failure	Progressively bloodier aliquots of bronchoalveolar lavage	Systemic manifestations such as fever, rash
Pathology	Diffuse alveolar damage	Diffuse alveolar damage	Diffuse alveolar damage
Pathogenetic drivers *	TNFα	Various cytokines	GM-CSF, G-CSF
Response to corticosteroids	Poor, some response after anti-TNF agents	Moderate response to high dose steroids	Excellent response
Prognosis	Favorable	Moderate to Poor	Very poor

* TNFα, tumor necrosis factor α; GM-CSF, granulocyte monocyte colony stimulating factor; G-CSF, granulocyte colony stimulating factor.

**Table 3 cancers-15-00680-t003:** Biomarkers associated with endothelial dysfunction related post-transplant syndromes.

Marker	VOD/SOS	TAM	IPS/DAH	GVHD
Coagulation factors				
VWF	↑	↑	-----	↑
TM	↑	↑	-----	↑
PAI1	↑	-----	-----	-----
Cell adhesion molecules				
ICAM	↑	-----	↑	↑
VCAM	-----	↑	↑	-----
E-selectin	↑	↑	-----	↑
P-selectin	↑	-----	-----	-----
Proinflammatory cytokines				
TNFa	↑	-----	↑	↑
IL6	-----	-----	-----	↑
sIL2R	↑	-----	-----	↑
Angiogenetic factors				
CEC	-----	-----	-----	↑
VEGF	↑	-----	-----	-----
Ang2	-----	-----	↑	↑
Panels				
EASIX	↑	↑	-----	↑
MAGIC (ST2, REG3a)	-----	-----	-----	↑

TM, thrombomodulin; PAI1, plasminogen activator inhibitor-1; ICAM, intercellular adhesion molecule; VCAM, vascular cell adhesion molecule; TNFa, tumor necrosis factor; IL6, interleukin-6; sIL2R, soluble IL2 receptor; CEC, circulating endothelial cells; VEGF, vascular endothelial growth factor; Ang2, angiopoetin-2; EASIX, endothelial activation and stress index; ST2, suppression of tumorigenicity 2; REG3A, regenerating islet-derived 3-alpha.

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
