# Peer review of "Endothelial Dysfunction Syndromes after Allogeneic Stem Cell Transplantation"

_cancers, 2023, doi:10.3390/cancers15030680_

Round 1

Reviewer 1 Report

In this very nicely written review article, an often neglected topic in stem cell transplantation is addressed in great detail.

It is a highly topical and relevant issue that will receive much attention in the peer group.

The current literature is considered, and recommendations for action in clinical practice can be derived. 

Overall, a successful and clear work that I can recommend for publication.

Author Response

We thank very much the reviewer fot the comments

Reviewer 2 Report

This is a well-conceived and written review article. Great work!

The authors presented a broad literature overview describing pathophysiology, pathogenesis and biomarkers of endothelial dysfunction.

I have no major concerns about this article. My minor comments and recommendations to the authors:

1.     Page 1, line 16-17: “Allo-HSCT remains the only therapy with curative potential for a variety of malignant and non-malignant hematological diseases”. I would recommend to delete hematological, because there are other non-malignant diseases that could be cured by HSCT (metabolic and immunodeficiency, not only hematological);

2.     References: number 1 and number 18 – the same reference “Luft, T.; Dreger, P.; Radujkovic, A. Endothelial cell dysfunction: a key determinant for the outcome of allogeneic stem cell 628 transplantation. Bone Marrow Transplant 2021, 56, 2326-2335. DOI: 10.1038/s41409-021-01390-y Please check and correct it and reorganize the list of references.

Author Response

We thank very much the reviewer for the useful comments

Response to comments:

1) Page 1, line 16-17: “Allo-HSCT remains the only therapy with curative potential for a variety of malignant and non-malignant hematological diseases”. I would recommend to delete hematological, because there are other non-malignant diseases that could be cured by HSCT (metabolic and immunodeficiency, not only hematological);

RESPONSE: The word ''hematological'' has been removed from the revised version of our manuscript

2) References: number 1 and number 18 – the same reference “Luft, T.; Dreger, P.; Radujkovic, A. Endothelial cell dysfunction: a key determinant for the outcome of allogeneic stem cell 628 transplantation. Bone Marrow Transplant 2021, 56, 2326-2335. DOI: 10.1038/s41409-021-01390-y Please check and correct it and reorganize the list of references.

RESPONSE: The mistake has been corrected. The references have been re-organized in the revised version of our manuscript 

Reviewer 3 Report

Endothelial cells are known to play a pivotal role in regulating coagulation and vascular homeostasis. Endothelial cells  injuries  after stem cell transplantation are responsible for a series of complications with high morbidity and mortality, which early recognition is crucial to improve outcome. 

In this paper the authors performed an exhaustive review of the pathogenesis and clinical presentation of the different endothelial dysfunction syndromes, and discussed the possible therapies and diagnostic tools.

The paper is clear, well planned, with updated references, and so tables and figures.

There are just few minor points to be clarified:

·       Page 5, line 216: table 1 refer to TA-TMA diagnostic criteria, not to that for SOS/VOD. 

·       Page 11, line 456: check abbreviation.

·       Section 3.3.2. There is evidence of a role of G-CSF administration in facilitating the development of diffuse alveolar hemorrhage?

·       Section 5. Atheromatosis: the authors described an increased risk compared to “not transplanted” population: how should it be managed?

Author Response

We thank very much the reviewer for the useful comments

1) Page 5, line 216: table 1 refer to TA-TMA diagnostic criteria, not to that for SOS/VOD. 

RESPONSE: We agree with the reviewer. This is a mistake. The word ''Table 1'' has been removed from the revised version of our manuscript. 

2) Page 11, line 456: check abbreviation.

RESPONSE: We agree with the reviewer. The correct is TA-TMA (transplant - associated thrombotic microangiopathy. This has been corrected also in the rest part of the paragraph and is marked in yellow.  

3) Section 3.3.2. There is evidence of a role of G-CSF administration in facilitating the development of diffuse alveolar hemorrhage?

RESPONSE: We agree with the comment. Actually a couple of DAH cases developed in healthy stem cell donors after G-CSF administration have been reported. This comment and the possible pathogenetic association of G-CSF with DAH have been added in the revised version of our manuscript (marked in yellow in section 3.3.2) 

4)  Section 5. Atheromatosis: the authors described an increased risk compared to “not transplanted” population: how should it be managed?

RESPONSE: Long term survivors after HSCT should be carefully followed by specialists for the early recognition and treatment of metabolic diseases such as diabetes, dyslipidemia, etc. Moreover a significant change into a healthy life style is mandatory for elimination of risk factors. These comments have been added in the revised version of our manuscript in section 5 (marked in yellow).